# Predicting Cardiovascular Rehabilitation of Patients with Coronary Artery Disease Using Transfer Feature Learning

**DOI:** 10.3390/diagnostics13030508

**Published:** 2023-01-30

**Authors:** Romina Torres, Christopher Zurita, Diego Mellado, Orietta Nicolis, Carolina Saavedra, Marcelo Tuesta, Matías Salinas, Ayleen Bertini, Oneglio Pedemonte, Marvin Querales, Rodrigo Salas

**Affiliations:** 1Faculty of Engineering, Universidad Andres Bello, Viña del Mar 2531015, Chile; 2Millennium Institute for Intelligent Healthcare Engineering (iHealth), Santiago 7820436, Chile; 3Biomedical Engineering School, Faculty of Engineering, Universidad de Valparaíso, Valparaíso 2362905, Chile; 4Health Sciences and Engineering Doctorate Program, Faculty of Medicine, Universidad de Valparaíso, Valparaíso 2540064, Chile; 5Center for Research and Development in Health Engineering (CINGS-UV), Universidad de Valparaíso, Valparaíso 2362905, Chile; 6Exercise and Rehabilitation Sciences Institute, School of Physical Therapy, Faculty of Rehabilitation Sciences, Universidad Andres Bello, Santiago 7591538, Chile; 7Fundación Cardiovascular Dr. Jorge Kaplan Mayer, Viña del Mar 2570017, Chile; 8Medical Technology School, Faculty of Medicine, Universidad de Valparaíso, Valparaíso 2540064, Chile

**Keywords:** cardiovascular rehabilitation, machine learning, joint distribution adaptation, transfer feature learning

## Abstract

Cardiovascular diseases represent the leading cause of death worldwide. Thus, cardiovascular rehabilitation programs are crucial to mitigate the deaths caused by this condition each year, mainly in patients with coronary artery disease. COVID-19 was not only a challenge in this area but also an opportunity to open remote or hybrid versions of these programs, potentially reducing the number of patients who leave rehabilitation programs due to geographical/time barriers. This paper presents a method for building a cardiovascular rehabilitation prediction model using retrospective and prospective data with different features using stacked machine learning, transfer feature learning, and the joint distribution adaptation tool to address this problem. We illustrate the method over a Chilean rehabilitation center, where the prediction performance results obtained for 10-fold cross-validation achieved error levels with an NMSE of 0.03±0.013 and an $R^{2}$ of 63±19%, where the best-achieved performance was an error level with a normalized mean squared error of 0.008 and an $R^{2}$ up to 92%. The results are encouraging for remote cardiovascular rehabilitation programs because these models could support the prioritization of remote patients needing more help to succeed in the current rehabilitation phase.

## 1. Introduction

Cardiovascular diseases represent the leading cause of death worldwide [1]. For 2019, 28,019 deaths due to circulatory system diseases were reported in Chile, representing 25.6% of the total deaths, with myocardial infarction and cerebrovascular disease being the most common events [2]. Hence, it is crucial to have an approach to cardiovascular patients to facilitate their rehabilitation.

Cardiovascular rehabilitation (CVR) represents the set of activities necessary to ensure that people with cardiovascular disease are in an optimal condition that allows them to occupy as usual a place as possible in society [3]. This involves restoring the biopsychosocial well-being of the subject, reintegrating them into their context without limitations, thus improving their clinical status, quality of life, and prognosis for survival. CVR phase I, or the hospital phase, is carried out after an acute cardiovascular event or surgery. Phase II has a maximum duration of 90 days and aims to recover or improve the patient’s physical and psychological capacity to the fullest. Finally, phase III seeks to maintain the objectives achieved in phase II, reintegrate the patient into daily life activities, and maintain healthy lifestyle behaviors (secondary prevention) [4].

As the phases were defined, a multidisciplinary team is required to achieve the optimal rehabilitation of cardiovascular patients, most of whom are located at the health center where the program is implemented. However, to contain the emergence of the coronavirus disease 2019 (COVID-19) pandemic, many centers have had to close spaces or limit non-emergency activities that are not linked to respiratory infection, thus affecting CVR since the number of sessions has been limited to minimize contact due to confinements and the probability of infection [5]. Therefore, remote monitoring is one way to provide continuity to rehabilitation programs without increasing the risk of patients and professionals acquiring COVID-19 infection.

These tools are gaining increasing support because they can overcome the logistical barriers of transportation and improve patient adherence and participation [6,7]. Although there are no longitudinal studies to determine the efficacy of this type of program, it has been observed that single or hybrid tools (sessions in centers together with remote monitoring) can also achieve optimal patient rehabilitation and not only in cardiovascular illnesses [8,9,10,11].

The progress made by artificial intelligence (AI) in recent years has facilitated the implementation of monitoring tools in cardiology [12]. This is because machine learning (ML) methods allow modeling highly complex combinations of variables [13,14]. A particular example is that CVR evaluation requires the study of clinical, psychological, and anthropometric metrics and those associated with risk factors and heart-healthy habits, among others. On the other hand, AI tools have allowed the integration of data from different sources, where sensors are used to capture variables such as blood pressure or heart rate; cell phones or web applications are also commonly used [12]. However, at this point, there is an issue to overcome: the type of sensors used in the patients at different rehabilitation centers are not the exact [15,16] makes the models not transferable between centers. Moreover, the data are usually limited and only sometimes available for all patients, even when they belong to the same center.

These challenges have been recently addressed in other areas. Authors from the British Ministry of Defense [17] proposed hierarchical learning to obtain good results to improve multiclassification algorithms through the subdivision of these algorithms into more general models when transfer learning is conducted on a dataset with limited data. To predict sub-feeder water uptake, the authors proposed a joint distribution adaptation (JDA)-based XGBoost transfer feature learning method [18]. Other authors have also proposed a transfer learning method with a transfer induction point (TIP) algorithm for data selection in large datasets to maintain transfer performance [19].

Therefore, this work presents a method for building a hybrid machine learning model that predicts if a patient ending phase II of a CVR will progress successfully to phase III. Due to the lack of prospective data compared with retrospective data, we built a model using only retrospective data with stacked machine learning first to obtain the cardiovascular risk predictor and then the rehabilitation probability. Transfer feature learning was applied to adapt the retrospective model to the new hybrid model, which included new features not available before COVID-19.

The main contribution of this research is oriented to two aspects: clinical and technological. First, this method represents a cardiovascular rehabilitation approach that could be used in the decision-making of the clinical staff involved. From the technical point of view, the process and its resulting models are the first in AI support for a clinical decision that seeks to address the problem of limited data by integrating two datasets without a coincidence in the number of features. This allows the aggregation of new variables through time that may arise from capturing data from new sensors.

## 2. Related Works

Several artificial intelligence models have been proposed for cardiovascular health monitoring or predicting cardiovascular diseases. Louridi et al. [20] presented an identification of cardiovascular diseases using ML. Working with the UCI heart disease dataset, the authors predicted the presence of cardiovascular disease. Similarly, Singh and Singh [21] built an ML model for classifying the presence of a cardiac arrhythmia. Segura et al. [22] proposed a methodology combining optimal feature selection methods with ML techniques for predicting cardiovascular disease.

Kántoch [23] used feature extraction and supervised ML algorithms to recognize sedentary behavior related to cardiovascular risk automatically. On the other hand, Fang et al. [24] used a trained neural network that takes extracted pulse wave features from a triboelectric textile sensor as inputs and generates two outputs representing the systolic and diastolic blood pressure. Additionally, addressing the issue of blood pressure monitoring, López et al. [25] applied ML models using data from a wearable device for acquiring blood pressure, heart rate, and number of steps, and also used a smartphone application and a web platform.

Recently, an ensemble of machine learning algorithms on lifestyle factors for cardiovascular risk prediction was presented by Huang et al. [26]. Other authors have used AI in CVR but for monitoring aspects such as adherence or motivation. Such is the case of Wallert et al. [27], who applied supervised ML models to investigate both established and novel predictors for internet-delivered cognitive behavior therapy (iCBT) adherence in patients with myocardial infarction and symptoms of anxiety, depression, or both (MI-ANXDEP). They used data from the multicenter Uppsala University Psychosocial Care Programme (U-CARE) [28] and applied a supervised machine learning procedure within a 3×10–fold cross-validated recursive feature elimination (RFE) resampling, which selected the final predictor subset that best differentiated adherers versus non-adherers.

Jahandideh et al. [29], on the other hand, developed a model able to predict individual intention to engage in outpatient CVR programs based on the first stage of the model of therapeutic engagement integrated into a socio-environmental context. The authors explored the effect of random forest-selected profile features on individual intention to engage in outpatient CVR. Tripoliti et al. [30] presented a platform for enabling heart failure patients to self-manage the disease and remain adherent while allowing specialists to monitor the patient’s health progress.

With the aim of monitoring patients in CVR programs, several systems combined with AI have been proposed. Desai et al. [31] presented a system for monitoring the health status of heart patients using machine learning and cloud computing called HealthCloud. This system predicts the presence of heart disease using data collected from blood tests, fluoroscopy, and electrocardiogram (ECG). Similarly, Alshurafa et al. [32] introduced Wanda-CVD, a smartphone-based remote health monitoring system designed to assist participants in reducing identified cardiovascular disease risk factors through wireless coaching using feedback and prompts as social support.

While assessing patients’ health status during rehabilitation, De Cannière et al. [33] evaluated whether a multi-parameter sensor (using a wearable ECG and accelerometer device) could be used during a standardized activity test to interpret functional capacity in the longitudinal follow-up of CR patients. The performance of ML models combining different features and using different kernel types was used to predict functional capacity, with promising results.

According to what was reviewed (summarized in Table 1), most of the existing research in the literature had a focus associated with the patient’s health status: blood pressure monitoring or the prediction of cardiovascular disease. In contrast, some others focus on adherence or motivation, and few predict rehabilitation.

## 3. Materials and Methods

### 3.1. Patients and Cardiovascular Rehabilitation Center

In this paper, data from patients of both prospective and retrospective studies from a cardiovascular rehabilitation center located in Viña del Mar, Chile, were used to develop the proposed model. Both datasets correspond to cardiovascular patients over 18 years of age, referred to comprehensive cardiovascular rehabilitation (Rehabilitation Center of the Dr. Jorge Kaplan Meyer Foundation), who have been diagnosed with (or undergone surgery for) acute myocardial infarction, heart failure, valvular failure, or coronary artery disease. Patients with any contraindication to physical exercise, including Parkinson’s disease, severe dementia, or psychiatric comorbidities that preclude initiation of the program were excluded. Phase II of the rehabilitation process of each patient covers approximately 90 days from the patient’s initial admission to discharge.

As retrospective data, information from 207 patients was used. In contrast, information from 20 patients was used as prospective data, whose measurements during rehabilitation generated more variables than those stored in the retrospective data. All the patients who participated in the prospective study signed an informed consent before participating. The Institutional Bioethical Committee approved this study.

### 3.2. Variables

#### 3.2.1. Retrospective Data

The retrospective dataset initially contained 278 variables, most of which had missing records. Therefore, a data cleaning process was carried out where 64 variables containing a sufficient number of study features were selected. Following the methodology proposed by [22], statistical techniques were then applied to reduce the number of variables further. In particular, principal components analysis, correlation analysis, and a logit model were used, in addition to the recommendations of clinical professionals regarding the evaluation of cardiovascular rehabilitation. This allowed us to select 22 variables, including clinical records from nutrition, kinesiology, and the psychological test SF-36 (summarized in Table 2).

#### 3.2.2. Prospective Data

Twenty new patients were monitored in their cardiovascular rehabilitation. In addition to the 22 variables already mentioned in the retrospective data, 22 additional variables (shown in Table 3) were registered through sensors, nursing evaluation, and blood tests (44 in total).

Blood tests were performed at a certified clinical laboratory, while nursing variables were obtained at the rehabilitation center. On the other hand, blood pressure variables were captured through a portable device (Holter) of pressure and pulse wave recording for arterial stiffness measurement (BR102 plus, Schiller). Similarly, echocardiographic data were collected using mobile 12-lead electrocardiography equipment (FD12 plus, Schiller). Finally, accelerometry data were obtained by installing a portable device (Actigraph wGT3X-BT) on the patient’s waist. The minimum time of use of the three instruments was 24 h.

Therefore, the selection of new variables corresponded to those that showed statistically significant differences between both measurements (evaluated through a Student’s *t*-test). In addition, these differences were evaluated through a logit model to see if they could predict the probability of rehabilitation. The final selection of the 22 variables (must be 22 or less not to exceed the original dataset) was obtained based on these criteria.

#### 3.2.3. Labeling of Patients

It was necessary to label each patient in the dataset and thus perform the training to predict the level of cardiovascular rehabilitation through an ML model. This process was performed by a set of clinical specialists who also added labels for the level of adherence and cardiovascular risk.

To this end, individual meetings were held with the health professionals of the rehabilitation center (kinesiologist, nurse, nutritionist, and psychologist) to obtain the details of the rehabilitation process on an individualized basis by professionals, with a focus on understanding the process from an interdisciplinary point of view. The main objective of these meetings was to learn about patient support tools and understand how each patient’s monitoring and control was carried out.

Subsequently, the professionals labeled an initial set of data. Each professional performed the task separately, delivering percentages of rehabilitation, treatment adherence, and cardiovascular risk from all existing patient information at the rehabilitation center that was made available to the professionals in an individualized PDF file per patient.

The percentages were distributed as follows:Level of cardiovascular rehabilitation and adherence:–0–25: Low level;–25–50: Medium-low level;–51–75: Medium-high level;–76–100: High level.Cardiovascular risk level (CVR):–0–25: Low CVR;–25–50: Medium-low CVR;–51–75: Medium-high CVR;–76–100: High CVR.

### 3.3. Stacked Machine Learning with Transfer Feature Learning

The machine learning proposal implies two phases. The first corresponds to the training of retrospective data with a model for predicting the probability of cardiovascular rehabilitation, using 24 variables (22 originals in addition to adhesion and cardiovascular risk labels). The second phase incorporates the prospective data (with 22 new variables) into the model trained with retrospective bases. For this, a reduction of dimensionality is applied to transform the information space of the new variables into a smaller space (retrospective data).

#### 3.3.1. Stacked Machine Learning Using Retrospective Data

This first phase of the proposal corresponds to predicting the probability of rehabilitation using retrospective data. It involves three stages (shown in Figure 1), starting with the data table corresponding to the difference between discharge and admission (Variable difference) and normalized before use.

Although a group of clinical specialists had previously analyzed the database and labeled the patients in terms of adherence and cardiovascular risk, there were cases in which this could not be done. Therefore, given the information available for those labeled, the first step was to predict adherence for cases in which this variable was absent. Similarly, in the second step, with the adherence variable absent, we estimated the cardiovascular risk for those patients who did not have this value.

For the construction of this risk model, previously, a sub-model was built to obtain a binary type risk prediction (see Figure 2). This classification sub-model aims to use its prediction as input for the final four-class cardiovascular risk model. This logic follows the hierarchical learning structure, where general classes are predicted first, and then the process continues with the predictions of subcategories or sub-classes of the same.

Once the labels for adherence and cardiovascular risk were completed, the third step consisted of predicting the probability of rehabilitation, the final output. The XGboost machine learning model was used to predict the probability of rehabilitation, adherence, and cardiovascular risk. The XGBoost (extreme gradient boosting) algorithm is a supervised learning technique that consists of a sequential assembly of decision trees (known as CART, the acronym for classification and regression trees). The trees are added sequentially to learn from the result of the previous trees and correct the error produced by them until such an error can no longer be updated (known as gradient descent) [34].

Other supervised machine learning models were applied to make comparisons, including gradient boosting, support-vector machine, random forest, and k-nearest neighbors.

#### 3.3.2. Transfer Feature Learning for Incorporating Prospective Data

The second phase merges retrospective data (22 variables) with data obtained from the new patients (22 additional variables). However, since there is a difference in the dimension of both spaces, an adaptation is required to incorporate the new variables and perform the training. For this purpose, given that there is already a pre-existing model trained with retrospective data, transfer feature learning techniques are proposed to take advantage of the existing model as a starting point in training a new model, which will also have unique characteristics.

For this transfer feature learning task, we chose the joint distribution adaptation (JDA) technique [35], a feature-based transfer type. This technique adapts the marginal or conditional distributions (in this project, we used the marginal distribution) to reduce dimensionality. It also integrates the maximum mean discrepancy (MMD) [36] with principal component analysis (PCA) to reconstruct a feature representation that matches the source and target domains.

The objective of JDA is to find an orthogonal matrix $A\in R^{mxk}$ such that the difference in both the marginal and conditional distributions is minimized. The objective function is defined as:(1)$$\min_{A^{T}XHX^{T}A}\sum_{c=0}^{C}tr\left(A^{T}XM_{c}X^{t}A\right)+\lambda {∥A∥}_{2}^{2}$$

According to the theory of constrained optimization, we can derive the Lagrangian function for the problem as:(2)$$L=tr\left(A^{t}\left(X\sum_{c=0}^{C}M_{c}X^{t}+\lambda I\right)A\right)+tr\left(\left(I-A^{T}XHX^{T}A\right)\Phi \right)$$

Obtaining $\frac{\partial L}{\partial A}=0$, then a generalized eigen-decomposition is obtained.
(3)$$\left(X\sum_{c=0}^{C}M_{c}X^{t}+\lambda I\right)A=XHX^{T}A\Phi$$

The latest prediction model follows the same procedure and training as the historical model. The only thing that changed is the model’s inputs, which correspond to the data transformed with JDA (Figure 3). This procedure delivers newly transformed data containing source and target domain features. Therefore, a model can be trained by taking advantage of the features of the existing model plus the new features of the most recent data.

In summary, we developed a stacked machine learning model (Figure 4), which includes transfer feature learning to adapt new to historical data for predicting cardiovascular risk levels. This label, with previous variables and the percent of adherence, allows for predicting the level of cardiovascular rehabilitation.

### 3.4. Model Explainability

Using the cardiovascular risk estimation model, an assessment was made of its explainability. It was possible to observe which variables contributed most to the risk and, therefore, to the probability of rehabilitation. The cardiovascular risk model was used because, for its construction, the original variables were considered but obtained from both the retrospective and prospective sets. In addition, a new variable has been added, named “binary risk”, which corresponds to the prediction of a cardiovascular risk submodel (see the hierarchical learning structure). The explainability was made using Shapley additive explanations (SHAP). SHAP is a game-theoretic approach to explaining the outcome of any machine learning model. It connects optimal credit allocation with local explanations using the classical Shapley values of game theory and their related extensions [37].

The classical Shapley value (Shapley values) is the average marginal contribution of an input parameter (feature) across all possible coalitions. SHAP is based on these Shapley values, where the objective is to interpret a model’s prediction by the contribution of each input parameter. For a model where the prediction function is f(x), and *F* is the set of all the input parameters (features) of the model, the Shapley values can be obtained as follows:(4)$$\phi_{i}=\sum_{S\subseteq F\setminus \{i\}}\frac{|S|!(|F|-|S|-1)!}{|F|!}\left[f_{S\cup \{i\}}\left(x_{S\cup \{i\}}\right)-f_{S}\left(x_{S}\right)\right]$$

In the above equation, |F| is the number of input parameters of the model, *S* is a subset of features that do not include the i-th feature, |S| is the cardinality of this subset, and fs() represents the prediction function of the model [37].

### 3.5. Performance Metrics

The study’s main objective was to predict the probability of rehabilitation in cardiovascular patients. For this reason, metrics associated with the regression task were used to evaluate the results by applying training through the 10-fold cross-validation method:Normalized mean square error (NMSE):
(5)$$NMSE=\frac{\sum_{i=1}^{n}{\left(Y_{i}-\hat{Y_{i}}\right)}^{2}}{\sum_{i=1}^{n}{\left(\hat{Y_{i}}\right)}^{2}},$$
where $Y_{i}$ is the vector of observed rehabilitation probability values and $\hat{Y_{i}}$ is the vector of predicted rehabilitation probability values.Mean absolute error (MAE):
(6)$$MAE=\frac{1}{n}\sum_{i=1}^{n}∥Y_{i}-\hat{Y_{i}}∥$$Mean absolute percentage error (MAPE):
(7)$$MAPE=\frac{1}{n}\sum_{i=1}^{n}∥\frac{Y_{i}-\hat{Y_{i}}}{Y_{i}}∥$$Coefficient of determination ($R^{2}$):
(8)$$R^{2}={\left[\frac{\sum_{i=1}^{n}(Y_{i}-\bar{Y_{i}})\left(\hat{Y_{i}}-{\tilde{Y}}_{i}\right)}{\sqrt{\sum_{i=1}^{n}(Y_{i}-\bar{Y_{i}}{)}^{2}}\sqrt{\sum_{i=1}^{n}{\left(\hat{Y_{i}}-{\tilde{Y}}_{i}\right)}^{2}}}\right]}^{2},$$
where ${\tilde{Y}}_{i}$ corresponds to the average of the predicted rehabilitation probability values.Spearman correlation coefficient (*r*):
(9)$$r=\frac{\mathrm{cov}(R\left(Y_{i}\right)R\left(\hat{Y_{i}}\right))}{\sigma_{R(Y_{i})}\sigma_{R(\hat{Y_{i}})}}$$
where R(·) corresponds to the ranks of the values, cov is the covariance and σ is the standard deviation. The results range in [−1,1].

## 4. Results and Discussion

The results for predicting cardiovascular rehabilitation are presented in Table 4. It is observed that using the machine learning model combined with the JDA strategy, a good prediction fit is obtained on average, reaching average values of 63% of $R^{2}$ and with low prediction errors as indicated by the NMSE, MAE, and MAPE.

Table 4 compares the proposed model with other methodologies. The XGBoost model performed better in all metrics, with the highest adjustment values and the lowest error rates. The models with the best performance were random forest and gradient boosting, while the models with the lowest performances were SVM and KNN.

It should be noted that the performance metric is shown as the average of the performance indicators. However, the best-achieved performance was an NMSE of 0.008 and an $R^{2}$ of up to 92%, as shown in Figure 5.

Making a direct comparison with the performance reported in the literature is complex, given that no predictive models were found for the probability of cardiovascular rehabilitation. However, the work of De Cannière et al [33] presented a machine learning model to predict functional capacity based on 6-min walking tests (6MWT) and used it as monitoring in cardiovascular rehabilitation, obtaining an MAE of 42.8 ± 36.8. Their result, in terms of error, was higher than that obtained in our work. However, different input variables, models (De Cannière et al. applied SVM), and output variables are being used.

Nevertheless, contrasting with those methods focused on patient monitoring, the main difference is the predictive model used. In the works of Desai et al. [31], Alshurafa et al. [32], or De Cannière et al. [33], models such as SVM, random forest, decision trees, or neural networks are used, while XGBoost is applied in our proposal. Moreover, none of the evaluated papers have faced the problem of limited data because, in this sense, our work is one of the first to show how to adapt two datasets with different dimensions. It is important to note that incorporating new variables from different sensors can improve prediction performance compared to using only retrospective data.

According to the explainability of the model’s results, SHAP showed the most influential variables in cardiovascular risk prediction. Among the ranking of SHAP variables (see Figure 6), many variables (and their importance) used by professionals can be observed when evaluating patients regarding the risk of cardiovascular risk. For example, for the classes ”low” (blue) and ”medium-low” (purple), the variable ”MET” has zero and low importance, respectively, according to SHAP. However, for the classes ”medium -high” and ”high” (represented in red and green, respectively), their importance is relatively high for predicting cardiovascular risk.

Additionally, with SHAP, it is possible to obtain the impact distribution of the variables. As seen in Figure 7, the *Y* axis shows the input variables of the model, and the *X* axis shows the SHAP values concerning the prediction of cardiovascular risk; a vertical bar is also shown on the right side as it goes from blue to red. This bar represents the magnitude of the variables (lower blue, higher red). With this graph, we can interpret how the variables impact the model in a specific way. For example, we can observe how the ”adherence”, when it obtains higher values (red), obtains lower SHAP values.

In contrast, when the ”adherence” is lower (blue), the SHAP values increase. This means that the model interprets that the greater the patient’s adherence to treatment, the lower their cardiovascular risk. On the other hand, nutritional variables, which present a more precise distribution of impact, are directly proportional to the SHAP values. The lower their values, the lower the SHAP values; these variables are as follows: ”Fat Mass”, ”Body Mass”, ”Weight”, ”ICC” and ”Visceral Fat”.

The main limitation of this research was the limited prospective data, whose number, although contemplated in the study design, was not initially intended to be so low. Health restrictions during the COVID-19 pandemic made it challenging to follow up with a more significant number of patients in the rehabilitation center. However, the good results obtained give reason to increase the number of individuals in this post-pandemic stage, which would increase the model’s performance.

## 5. Conclusions

This work proposed a new stacked machine learning model for predicting the cardiovascular rehabilitation of patients with coronary artery disease by applying transfer feature learning using a set of retrospective and prospective data. Our approach reaches a low normalized mean square error of 0.030±0.013 and an adjustment of 92%, obtaining a better fit in comparison to different ML models used on similar data, suggesting that combining datasets with similar variables using techniques such as JDA can allow the further use of retrospective data in combination with newer features to make a good rehabilitation prediction model. Additionally, using SHAP, we evaluated the explainability of the model, showing that variables such as high adherence tend to reduce risk cardiovascular level and, hence, increase the probability of rehabilitation.

## Figures and Tables

**Figure 1 diagnostics-13-00508-f001:**
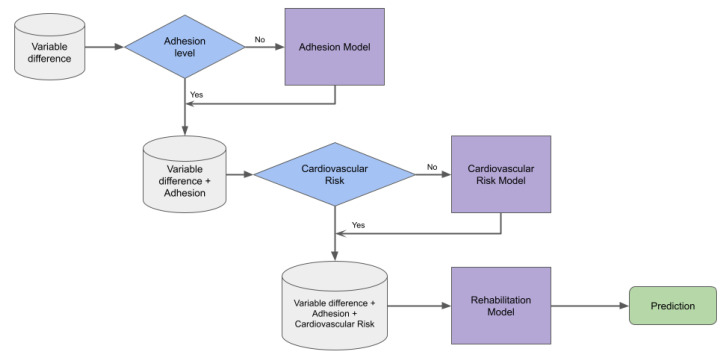
Proposal model for predicting the probability of rehabilitation using the retrospective cardiovascular data. It involves three stages: predicting adherence and cardiovascular risk for those patients who did not have these values, and finally predicting rehabilitation with the same machine learning approach.

**Figure 2 diagnostics-13-00508-f002:**
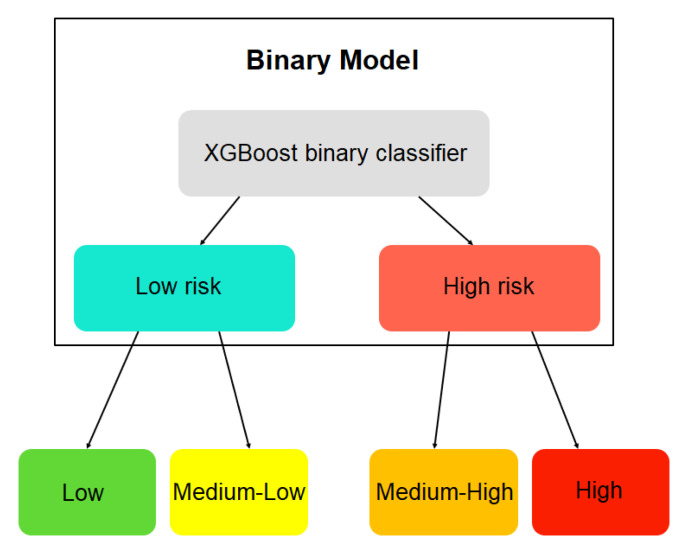
Scheme of the hierarchical learning model for cardiovascular risk.

**Figure 3 diagnostics-13-00508-f003:**
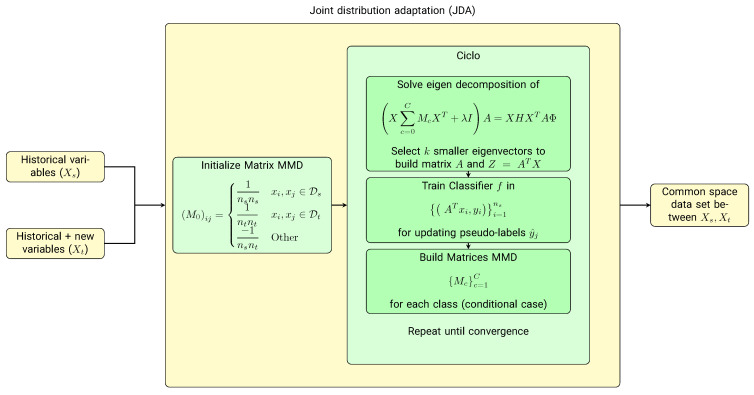
Transfer feature learning scheme for incorporating new variables into the cardiovascular rehabilitation model. Given that there is a difference in the number of variables in the retrospective database and those obtained for the new patients, a transfer feature learning algorithm based on dimensional reduction, the joint distribution adaptation (JDA), is proposed to combine the feature spaces of both sets of variables.

**Figure 4 diagnostics-13-00508-f004:**
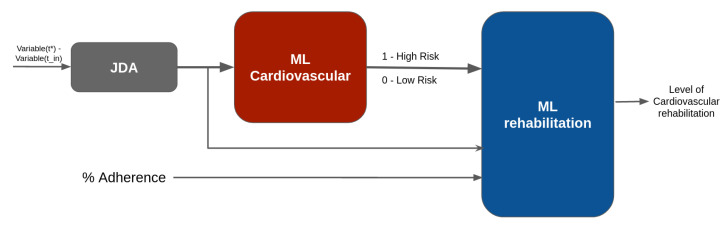
Stacked machine learning with transfer feature learning.

**Figure 5 diagnostics-13-00508-f005:**
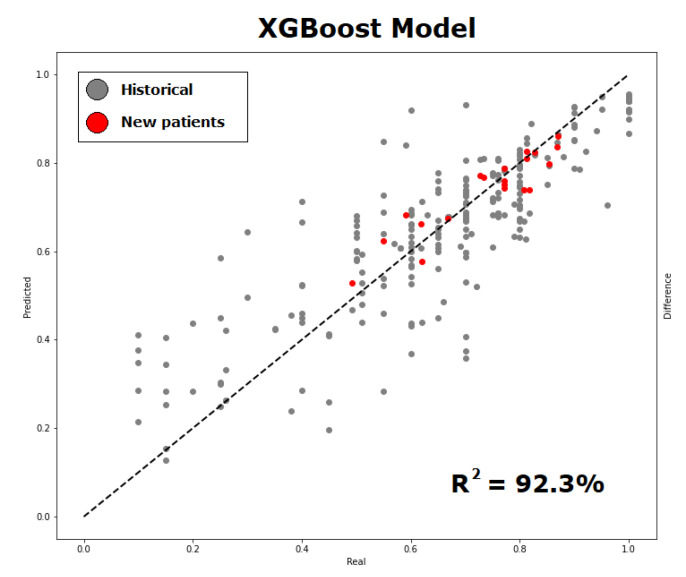
Probability of cardiovascular rehabilitation (predicted versus observed values) using the proposed methodology of incorporating new variables into the retrospective data, observing that new data (in red) contributes to a good adjustment of prediction. The graph shows the best performance result obtained.

**Figure 6 diagnostics-13-00508-f006:**
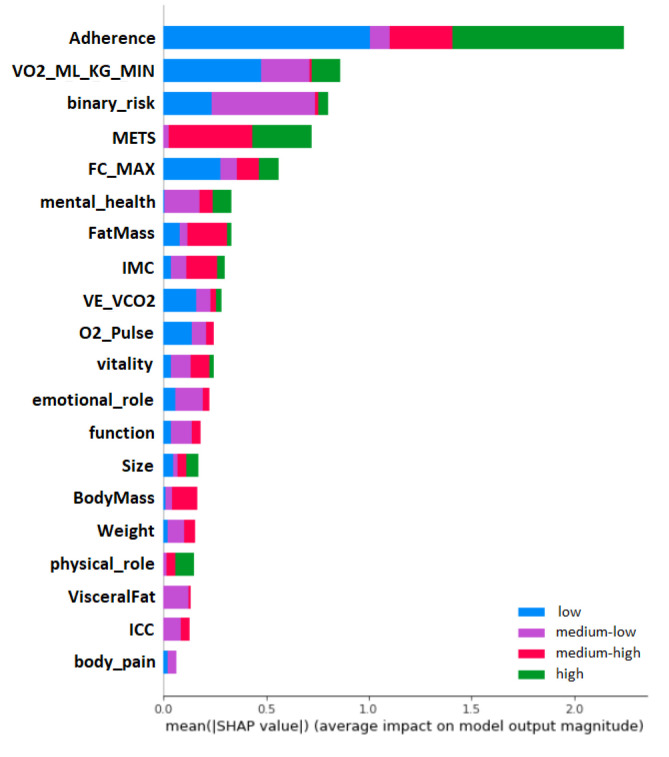
Global impact of the variables on the cardiovascular risk model. It can be observed that adherence, oxygen uptake and MET have a major impact on the prediction, allowing to reduce the cardiovascular risk.

**Figure 7 diagnostics-13-00508-f007:**
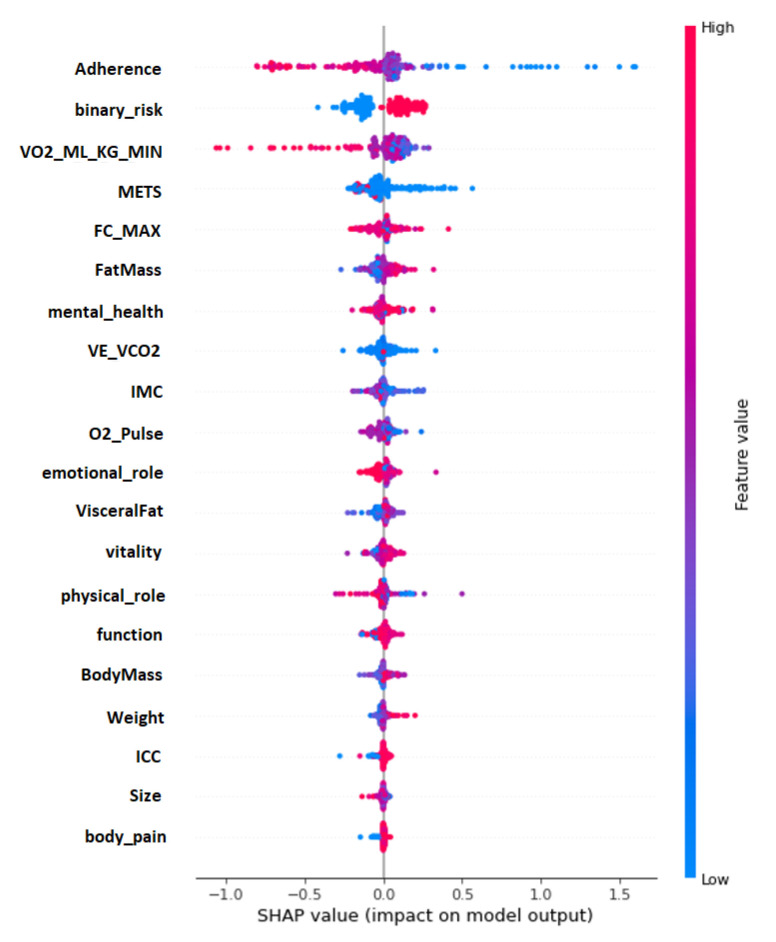
Impact distribution of the variables in the cardiovascular risk model.

**Table 1 diagnostics-13-00508-t001:** Summary of some of the state-of-the-art models used in AI support for cardiovascular health.

Author	Model Description
Louridi et al. [20]	Naive Bayes.
Singh and Singh [21]	Random forest.
Huang et al. [26]	Naive Bayes, random forest and support-vector classifier.
Kántoch [23]	Binary decision trees, discriminant analysis model, naive Bayes, k-nearest neighbors classification, support-vector machines and artificial neural networks.
Fang et al. [24]	Neural Networks.
López et al. [25]	Random forest, decision tree, support-vector regression, Bayesian ridge, linear regression, and polynomial regression.
Wallert et al. [27]	Random forest.
Jahandideh et al. [29]	Ordinal logistic regression and random forest.
Desai et al. [31]	Support-vector machine, k-nearest neighbors, neural networks, logistic regression, and gradient boosting trees.
Alshurafa et al. [32]	Logistic regression, C4.5 decision trees, k-nearest neighbors and naive Bayes.
De Cannière et al. [33]	Support vector machine.
Tripoliti et al. [30]	Random forests, logistic model trees, J48, rotation forest, support-vector machines, radial basis function network, Bayesian network, nive Bayes.

**Table 2 diagnostics-13-00508-t002:** Variables from the retrospective data used to develop the proposed model.

Kinesiology	Nutrition	Psychology
Oxygen uptake (L/min)	Weight (kg)	Function (score 0–100)
Oxygen uptake (mL/Kg/min)	Height (meters)	Physical role (score 0–100)
Maximum heart rate (beat/min)	Body mass index (kg/m^2^)	Bodily pain (score 0–100)
O_2_ pulse (mL/beat)	Waist-hip index	General health (score 0–100)
Ventilation/CO_2_ production ratio	Lean mass (%)	Vitality (score 0–100)
Metabolic Equivalent Task (MET)	Body mass (%)	Social function (score 0–100)
	Fat mass (%)	Emotional role (score 0–100)
	Visceral fat mass (%)	Mental health (score 0–100)

**Table 3 diagnostics-13-00508-t003:** Additional variables from new patients for developing the proposed model.

Source	Variable
Pressure Holter	Overall mean systolic, overall mean diastolic, overall mean heart rate, overall mean blood pressure, overall mean pulse pressure.
Accelerometry	Kilocalories, step count, MET, total moderate to vigorous physical activities (MVPA), Borg strength, resting heart rate.
Blood test	Glycemia, total cholesterol.
Nursing evaluation	Description of the prescribed diet, description of the prescribed medication, identification of the disease process, and description of the prescribed activity.
Echocardiogram	Tricuspid annular plane systolic excursion (TAPSE), fractional shortening (FS), ejection fraction (EF), stretch and shortening measure (A).

**Table 4 diagnostics-13-00508-t004:** The 10-fold cross-validation performance results of the proposed stacked machine learning model for predicting cardiovascular rehabilitation.

Base Models	NMSE	$R^{2}$	*r*	MAE	MAPE
Random forest	0.0354 ± 0.011	0.562 ± 0.109	0.749 ± 0.115	0.092 ± 0.016	0.252 ± 0.094
XgBoost	0.030 ± 0.013	0.630 ± 0.189	0.760 ± 0.162	0.086 ± 0.021	0.212 ± 0.120
Gradient boosting	0.037 ± 0.011	0.525 ± 0.119	0.714 ± 0.107	0.094 ± 0.017	0.257 ± 0.094
KNN	0.063 ± 0.018	0.209 ± 0.088	0.462 ± 0.079	0.125 ± 0.020	0.357 ± 0.126
SVM	0.059 ± 0.015	0.232 ± 0.137	0.456 ± 0.091	0.126 ± 0.019	0.312 ± 0.104

## Data Availability

We do not have authorization from the ethical committee to publicly release the data.

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
