# Peer review of "Predicting Cardiovascular Rehabilitation of Patients with Coronary Artery Disease Using Transfer Feature Learning"

_diagnostics, 2023, doi:10.3390/diagnostics13030508_

Round 1
Reviewer 1 Report
The study conducted by authors is very interesting and contemporary. The value of presented method for building a cardiovascular rehabilitation prediction model using using machine learning to address this problem could not be overestimated.
Line 83-86. There is no need to specify forthcoming paragraphs in this section of the manuscript
I believe that introduction part of the text could be shorten. Please, leave the relevant and specific information only and avoid unnecessary description. For example, there is Table 1 that summarizes the main points in section 2. Related works. Do not repeat this in full in main text.
Are 20 patients of the prospective study sufficient to build the prediction model? I think that this number of the patients is not of great importance to inputting in this model.
Author Response
The authors appreciate the time spent by the reviewers to provide comments and suggestions that allow us to improve our work. In what follows, we answer each of the given observations of the reviewers.
1. The study conducted by authors is very interesting and contemporary. The value of presented method for building a cardiovascular rehabilitation prediction model using machine learning to address this problem could not be overestimated.
Answer: Thank you very much. Our purpose is to address the problem of cardiovascular rehabilitation and generate tools for clinical assistance in this field.
2. Line 83-86. There is no need to specify forthcoming paragraphs in this section of the manuscript.
Answer: Thanks for the suggestion and observation. We have dropped these lines in the document.
3. I believe that introduction part of the text could be shorten. Please, leave the relevant and specific information only and avoid unnecessary description. For example, there is Table 1 that summarizes the main points in section 2. Related works. Do not repeat this in full in main text.
Answer: We appreciate the observation. We have summarized both sections (introduction and related works), reduced 16 lines (around two paragraphs), and highlighted the main ideas.
4. Are 20 patients of the prospective study sufficient to build the prediction model? I think this number of the patients is not of great importance to inputting in this model.
Answer: thanks for this observation. We fully agree that the more data, the better the performance of machine learning models. However, it is challenging to have enough data in clinical studies (and even more so in monitoring). Hence, part of the objective of the study (as described in the final paragraph of the introduction, between lines 74 and 80) is to develop a model that allows the prediction of the probability of cardiovascular rehabilitation but using limited data as it is closer to the reality of rehabilitation centers. Also, the methodology section described that we used retrospective data for training, adding new variables (from prospective data) to improve the model.
Reviewer 2 Report
Dear Editor and Authors,
I read the paper entitled 'Predicting cardiovascular rehabilitation of patients with coronary artery disease using transfer feature learning' with great interest.
This paper concerns actual topic and presents a method for building a cardiovascular rehabilitation prediction model using retrospective and prospective data with different features, using hierarchical learning, transfer learning, and the joint distribution adaptation tool to address this problem. The title describes the core message of the paper. The abstract incorporates key messages, in a concise manner. The structure of the paper is accurate. Importantly, the results are encouraging for remote cardiovascular rehabilitation programs because these models could support the prioritization of remote patients needing more help to succeed in the current rehabilitation phase.
However, I have same suggestions regarding this paper.
1. There is no need to expand abbreviations more than one time, e.g. Shapley Additive exPlanations (SHAP).
2. Captions under the figures are not very clear in some places, e.g. Figure 5 and 6.
3. Please add some paragraph about limitations of this study.
Author Response
The authors appreciate the time spent by the reviewers to provide comments and suggestions that allow us to improve our work. In what follows, we answer each of the given observations of the reviewers.
-
There is no need to expand abbreviations more than one time, e.g. Shapley Additive exPlanations (SHAP).
Answer: Thanks for this suggestion. We corrected the abbreviations throughout the text.
-
Captions under the figures are not very clear in some places, e.g. Figure 5 and 6.
Answer: Thanks for this observation. These captions were edited.
-
Please add some paragraph about limitations of this study.
Answer: Thanks for this suggestion. This paragraph was added and can be observed between lines 341 and 346.